# Automated Animal Training and Iterative Inference of Latent Learning Policy

## Abstract

Progress in understanding how individual animals learn requires high-throughput standardized methods for behavioral training and ways of adapting training. During the course of training with hundreds or thousands of trials, an animal may change its underlying strategy abruptly, and capturing these changes requires real-time inference of the animal's latent decision-making strategy. To address this challenge, we have developed an integrated platform for automated animal training, and an iterative decision-inference model that is able to infer the momentary decision-making policy, and predict the animal's choice on each trial with an accuracy of 80%, even when the animal is performing poorly. We also combined decision predictions at single-trial resolution with automated pose estimation to assess movement trajectories. Analysis of these features revealed categories of movement trajectories that associate with decision confidence.

## 1 Introduction

Learning – the change of neural representation and behavior that results from past experience and the consequences of actions – is important for animals' survival and is a central topic in biological intelligence [1]. Different individuals may apply different strategies to the learning process, reflecting their individual biases [2–4]. Indeed, substantial differences in sensory biases, locomotion, motivation, and cognitive competence have been observed in populations of fruit flies, rodents and primates. Thus, it is critical to investigate learning at the individual level.

Rodents, especially the mouse, have become popular experimental animals for studying associative learning and decision-making, due to the widely available transgenic resources [5]. They can learn to perform complex decision-making tasks that probe cognitive components such as working memory and selective attention [6–8]. However, differences in learning strategies across individuals have rarely been addressed, partly owing to the limitations of data gathering and methods for data analysis.

Studying differences among individuals requires training and collecting data from multiple animals in a standardized and high-throughput fashion. The training procedures are often time-consuming, requiring several days to many weeks, depending on the task [9, 10]. Human intervention in this process causes additional variability, therefore an automated animal training procedure is preferential for the purpose. In addition, analysis approaches tend to focus on the averaged performance over many trials, while changes in behavior may deviate at any trial, and thus necessitating modeling behavior at the single-trial level.

To address these challenges, we developed an integrated platform for automated-training of group-housed mice and analysis of behavioral changes in learning a decision-making task. We designed hardware that makes use of implanted radio frequency identification (RFID) chips to identify each mouse, and guides the animal into a behavior training box. Synchronized video recordings and

Submitted to 33rd Conference on Neural Information Processing Systems (NeurIPS 2019). Do not distribute.

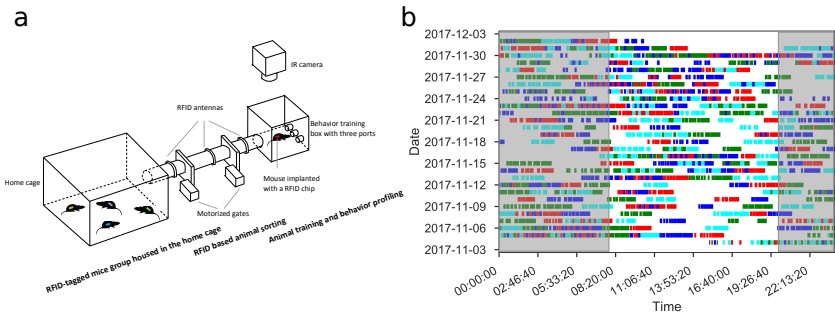

Figure 1: Automated training system and its performance. (a) Schematic of the automated training system. (b) Activity trace of a cohort of five animal in the behavior box, colorcoded by animal.

decision-making sequences were acquired during animal learning. To analyze the decision-making sequences, we developed an iterative generalized linear model, which predicts the animal's choice in each trial, and is updated immediately based on the animal's actual choice. To analyze the animal's behavior during the task in greater detail, we extracted the animal's pose from video data. We built an model on these extracted features to predict the animal's decisions. Interestingly, we found pre-stimuli behavior were predictive of the correctness of the animal's choice.

## 2 Automating Tasks and Animal Identification

We designed our automated training system as follows (Fig. 1a): RFID-tagged mice are grouped in a common home cage where food and bedding is supplied. The home cage connects to a behavior training box through a gated tunnel. The gates are controlled by a RFID animal sorting system. The behavior box is outfitted with three ports, each of which contains a photo-transistor to detect snout entry, a solenoid valve to deliver water reward, and a light emitting diode (LED) to present visual cues. The entire apparatus is orchestrated by a master program that coordinates the RFID sorting device, the behavioral box control system, synchronized video recording, data management and logging. We tested the automated RFID sorting and animal training system by training group-housed mice to learn a variety of decision tasks. The training period lasted 28 days, with up to five mice in the common home cage. Each animal occupied the training box for 3-4 hours per day throughout the entire training period (Fig. 1b). Each animal was trained for over 900 trials (10 sessions)

## 3 Online Latent Strategy Inference

We developed a generalized linear model (GLM) to map decision policies relevant to the animal's decision-making to its choices through logistic regression. Because a change of policy can happen at any trial during learning, we developed the model to make online trial-by-trial choice predictions based on various strategies the animal might plausibly use. The model works in an iterative two-step process. In the prediction step, the model makes a prediction for the next decision based on the input factors. Once the outcome of the animal's decision is observed, an error term between the model's prediction and the observation is computed. This error, after weighting by a reward factor and a temporal discount factor, is fed back to the loss function. In the update step, the model is updated by minimizing the regularized loss function. The temporal discount factor accounts for the possibility that the most recent trials impact the current decision more than remote trials. The reward factor accounts for the fact that water rewards and timeout punishments may have effects of different magnitude on the updates of the animal's policy (Fig. 2).

We illustrate the utility of this model by fitting results from a binary visual task, in which one of the two side ports lights up to indicate the location of the reward, and the optimal policy is to simply poke the port with the light. The GLM makes a prediction for the outcome of each trial based on a weighted combination of several input variables: the current visual stimulus, a constant bias term, and three terms representing the history of previous trials. These inputs from a previous trial include the port choice, whether that choice was rewarded, and a term indicating the multiplicative interaction between the choice and reward (Choice x Reward). This term supports a strategy called

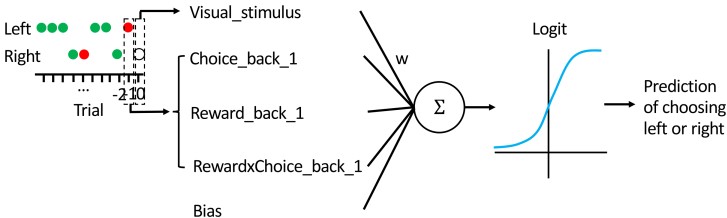

Figure 2: Illustration of the iterative inference model used to infer policies for the visual discrimination task. The model's prediction is based on the output of a logistic function whose input is the weighted sum of a visual stimulus term, a bias term, and three history-dependent terms. The stimulus can be on the left or right and the choice can be rewarded (consistent with the stimulus, indicated by a green dot) or unrewarded (opposite to the stimulus, indicated by a red dot).

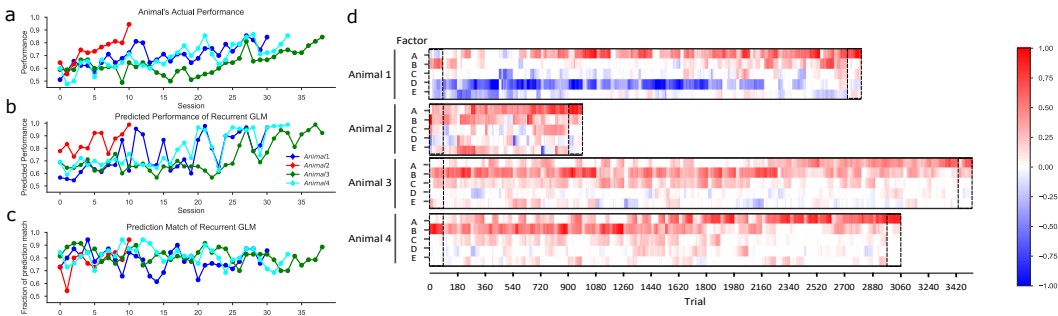

Figure 3: GLM performance. (a) The animal's actual performance over time in the visual task. (b) Performance as predicted by the GLM. (c) Fraction of choices predicted correctly by the GLM. (d) Interpretation of policies during learning: Policy matrices recovered for the four animals show distinct individual learning processes. Dashed rectangles highlight the first and last sessions of each animal

win-stay-lose-switch (WSLS), which chooses the same port if it was rewarded previously and the opposite one if not. Predictions from the iterative GLM matched 80% of the animals' actual choices (Fig. 3c), and the predicted accuracy of each animal captured the actual fluctuations of its learning curve (Figs. 3a and 3b). In addition, the iterative GLM serves to infer what policy the animal follows in making decisions, reflected by the weight of each input term. By following this weight vector across trials one obtains a policy matrix that documents how the animal's policy changes during learning (Fig. 3d).

# 4   Behavior Analysis

While predicting choice by inferring the underlying policy is important for training animals on a timescale of several trials, it would be more beneficial to further predict choice through pre-trial behavior. When a subject is completing a trial, momentary lapses in concentration while information related to this trial is presented can adversely affect the decision in the next trial. To explore this hypothesis, we used DeepLabCut [11] to train a pose estimator on the body landmarks, including nose, centroid, and tail-base. We fitted the pre-stimuli behavioral pose of the animal to the animal's choice accuracy through a boosting decision tree classifier, allowing us to decode whether the animal would make a correct decision. We show the k-fold validation accuracy for several animals (Fig. 4a). We find that independent of the type of choice, we can predict the correctness with above 80% accuracy. Analyzing the behavioral trajectories, we find that behaviors that suggest hesitation are captured in this model (Fig. 4b-d).

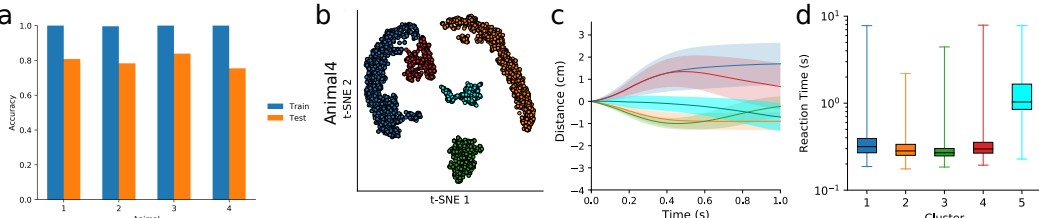

Figure 4: (a) Training and k-fold validation performance of boosting decision tree classifier trained on 6 seconds of pre-stimulus behavior pose data. (b) t-SNE of individual trial behaviors, grouped into 5 clusters. c) Visualization of the behavior traces along the behavior-box indicate three distinct groups of movement patterns, including a cluster for hesitation. d) Reaction plot of the 5 clusters illustrate the difference in reaction time.

## 5 Conclusion

We've presented a suite of hardware and machine learning tools for the automation of neuroscience experiments. Our method allows us to test multiple animals for thousands of trials without any human intervention. The policy inference model predicts an animal's choices with an accuracy of 80%. And the choice prediction model using pre-stimuli behavioral trajectories informs whether the animal would make correct decisions in the future. Altogether, this method can be extended to allow adaptive training that would further reduce the training time to get to the desired behavior for neural recordings.

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
