# OpenReview forum: "Automated Animal Training and Iterative Inference of Latent Learning Policy"
_NeurIPS.cc/2019/Workshop/Neuro_AI — Submitted to Real Neurons & Hidden Units @ NeurIPS 2019_

### Official Review · AnonReviewer1 · 2019-09-24
**Interesting setup to understand individual animals but more convincing analyses and results are needed**

**Clarity:** 3

**Comment:**

The hardware setup is good and the direction of understanding individual animals is interesting. However, more rigorous tests and analyses are needed. It would be also good to see how these understandings can improve the animal training.

**Category:**

AI->Neuro

**Clarity Comment:**

In figure 2, the dashed boxes on the left overlap with the number. In figure 3(d), the factors A,B,C,D,E are a little bit confusing and should be consistent with previous explanation. In general the figures are understandable.

**Evaluation:**

2: Poor

**Importance:**

3: Important

**Importance Comment:**

It is interesting to come up with an automated animal training platform to understand animals individually. They use generalized linear model to learn the decision-making online and also analyze how pre-trial movement trajectories could affect the decision-making. The idea is potentially useful but the current model and tested task are basic.


**Intersection:**

2: Low

**Intersection Comment:**

While the authors use machine learning techniques to model and analyze decision-making policy of individual animals, the inference model is based on the animal’s choice and movement trajectory instead of the activity of real neurons.

**Rigor Comment:**

In line 51, the authors claim they test the mice to learn a variety of decision tasks but only report a simple binary visual task and doesn't mention how to generalize to more complicated task. There was little intuition and without any explanation to show why GLM is preferred to other methods. Also, since there is no comparison to other methods at all, it is not clear that 80% accuracy is high or low for a binary visual task.

**Technical Rigor:**

2: Marginally convincing

---

### Official Review · AnonReviewer2 · 2019-09-26
**A decent neuroscience paper, but where’s AI?**

**Clarity:** 3

**Comment:**

Although the preliminary results presented here is not particularly novel nor significant judging from the manuscript, the experimental protocol they set up is quite neat, and I hope the authors will address interesting questions using this system.

**Category:**

AI->Neuro

**Clarity Comment:**

Fig. 4 c-d are not clear. What does the distance (y-axis of Fig. 4c) mean? And how can we tell that the cyan cluster correspond to hesitation, not inattention?

**Evaluation:**

2: Poor

**Importance:**

3: Important

**Importance Comment:**

In this paper, the authors developed an automated training process for mice, and analysed the decision-making behaviour by training GLM on decision related variables, and a classifier on outputs from Deeplabcut. Automated training of rodents is now practiced in more than several laboratories (eg. Winter & Schaefer, 2011; Poddar et al., 2013), and the behavioural analysis done here is pretty standard too, so the novelty of this work is rather limited.

**Intersection:**

1: Very low

**Intersection Comment:**

Behavioural analysis was done with popular methods for which no contribution has done from this particular work. Because of that, I think the level of intersection is quite low.

**Rigor Comment:**

It is probably just a clarity issue, but it is unclear whether the 80% performance they observed in section 4 is really significant from the manuscript alone. When 80% of the trials are the correct trials, even without training any classifier, you can trivially predict the performance with 80% chance. The authors should mention how the bias was controlled.

**Technical Rigor:**

4: Very convincing

---

### Official Review · AnonReviewer3 · 2019-09-27
**Interesting premise, but results are not there yet**

**Clarity:** 2

**Comment:**

Focusing on the methods and results of the behavioral analysis as compared to the more common GLM analysis would strengthen the paper - it is an interesting premise that deserves to be better explored.

**Category:**

AI->Neuro

**Clarity Comment:**

The methods and result sections are very unclear. For the purposes of this conference, there is a focus on the two types of models to predict the choice in the results section, but the focus of the other sections was to build the automating of tasks and animal identification - it is unclear how these two tie together. How exactly would the prediction of the animal's choice using either the GLM or the behavior recording help in adaptive training? Moreover, the models have obviously been built post-hoc.

That being said, there is potential in this analysis- if the authors concentrate on the prediction of the mouse's choice using (a) stimulus and (b) movement parameters, and make statements on how the behavior gives us further information about the mouse's future choice than just the stimulus and previous choices, then this paper could have a nice message.
Tightening up the message of the paper and focusing on the results of the modeling effort, and explaining these results a lot better, as opposed to spending valuable space detailing the automating of the rigs, would go a long way.

1) What is A,B,C,D,E in Figure 3d? - any interpretability in this figure would be great.
2) What do the different clusters correspond to in Figure 4, apart from 'hesitation'?
3) What is the 'distance' (cm) in Figure 4c?
4) How does the accuracy in Figure 4 compare to the accuracy values in Figure 3?
5) A lot more information about Figures 3 and 4 is needed.


**Evaluation:**

3: Good

**Importance:**

3: Important

**Importance Comment:**

This paper analyzes the ability to predict of a mouse's choice in a self-initiated 2AFC. Two different types of models are studied - a GLM with various terms including current stimulus, as well as a model with bodyparts tracked using DLC.
While the models used are interesting, and the potential behavioral result is interesting if true, there should be a lot more analysis on the results. At this stage, the findings are currently uninterpretable and unclear. Description of methods is lacking.

**Intersection:**

3: Medium

**Intersection Comment:**

A variety of machine learning tools are used in this study, from DeepLabCut to t-SNE to a boosting classifier. Not sure if that is enough for 'intersection'.

**Rigor Comment:**

The methods section for the behavioral analysis (Section 4) really needs to have more details in it for us to judge the importance of the result. A GLM is more commonly used for predicting future choice. If indeed the movements are offering much more insight than the GLM, especially the GLM without the stimulus information (easier task), then it's an interesting result. There is no comparison of the behavioral model to anything else currently, including chance levels.

How much more can the behavioral model capture than the GLM model, if any? Was this run on all trials or just a subset of the trials where the mouse was performing at or above 80% anyway?

Why was the behavioral model performing as well as it was? If the authors used all the time right up to the stimulus and the mouse didn't move between the stimulus and the choice, the mouse would already be at the chosen port and it would be easy to tell the choice.



**Technical Rigor:**

3: Convincing

---

### Decision · Program_Chairs · 2019-10-01

**Decision:**

Reject

**Comment:**

Unfortunately, we had more submissions than we could accept and based on the review process, we have decided not to accept your submission.  Nevertheless, thank you for your submission and interest in our workshop.